# Women's postpartum family planning use and intention in Ethiopia: Disparities in the agrarian and pastoral contexts from a community-based cross-sectional study

Agumasie Semahegn [1]*, Gizachew Tadele Tiruneh[2], Alemnesh Hailemariam Mirkuzie[2], Omar Mohammed[1], Nebreed Fesseha[2], Shegaw Mulu[1], Wubegzier Mekonnen[3], Addis Girma[1], Chala Tesfaye[2], Mikiyas Teferi[1], Biruk Bogale[2], Hillina Tadesse[2], Derbe Tadesse Abate[1], Meskerem Abebaw[1], Mebrie Belete[1], Miftah Yasin[1], Netsanet Belete[2], Zemzem Mohammed[4], Lidiya Tefera[5], Frank DelPizzo[5], Yibeltal Kifle Alemayehu[6,7], Abdulhalik Workicho[4,8], Muluken Dessalegn Muluneh[1], Addis Tamire[1], Temesgen Ayehu[1], Dessalew Emaway[2], Misrak Makonnen[1], Mesele Damte Argaw[1]

**1** Amref Health Africa in Ethiopia, Addis Ababa, Ethiopia, **2** JSI Research & Training Institute, Inc, Addis Ababa, Ethiopia, **3** School of Public Health, College of Health Sciences, Addis Ababa University, Addis Ababa, Ethiopia, **4** Ministry of Health, Addis Ababa, Ethiopia, **5** Gates Foundation, Ethiopia Country Office, Addis Ababa, Ethiopia, **6** MERQ Consultancy P.L.C, Addis Ababa, Ethiopia, **7** Department of Health Policy and Management, Faculty of Public Health Institute of Health, Jimma University, Jimma, Ethiopia, **8** Fenot Project-Harvard T.H. Chan School of Public Health in Ethiopia, Addis Ababa, Ethiopia

* agumas04@gmail.com

## Abstract

Despite the substantial improvement on maternal health service use and impactful intervention to reduce the risk of maternal and child deaths, uptake of Postpartum Family Planning (PPFP) has remain a critical gap in Ethiopia. There is a paucity of context-based evidence is crucial in addressing these gaps to inform policies. We assessed women's current use and intention on postpartum contraception in the agrarian and pastoral contexts of Ethiopia. A community-based house-to-house survey was conducted in ten selected Woredas from the agrarian and pastoral contexts of Ethiopia, for baseline assessment to design a community-based lifesaving package delivery model. Data were collected among randomly selected 3,097 women using structured web-based tool and analyzed using Stata/SE 18.0. A multi-level mixed-effect logistic regression was used to identify factors associated with women's current PPFP use and intention for future use in the agrarian and pastoral contexts. Only a quarter of women (25.3%, 95%CI:23.8%-26.9%) used PPFP with significant variations in agrarian (60.6%) and pastoral (0.9%) contexts. More than one-third (37.5%, 95%CI:35.8-39.2%) of women had the intention to use modern PPFP. Factors influenced both current and future PPFP use include women's antenatal care visits (AOR:3.46; 95%CI:2.25-5.32), strong social support (AOR:1.75; 95%CI:1.23-2.49), autonomy on FP use (AOR:3.25; 95%CI:1.89-5.59), and favorable

**Data availability statement:** All relevant data are within the paper and its Supporting Information files.

**Funding:** This study was financially supported by the Gates Foundation (INV-002643 to MM and INV-037995 to DE). The funders had no role in study design, data collection and analysis, decision to publish, or preparation of the manuscript.

**Competing interests:** The authors have declared that no competing interests exist.

**Abbreviations:** AIC, Akaike's information criterion; AOR, Adjusted Odds Ratio; CI, Confidence Interval; DIC, Deviance information criterion, HCP, Healthcare providers; HEW, Health Extension Workers; ICC, Intra-class correlation coefficient; PPFP, postpartum family planning; SE, Standard error; WHO, World Health Organization.

attitude towards equitable gender norms (AOR:1.51; 95%CI:1.12-1.71). Nevertheless, women who had no access to health facilities (AOR:0.70; 95%CI:0.49-0.99), history of home birth (AOR:0.53; 95%CI:0.39-0.72), being from pastoral communities (AOR:0.03; 95%CI:0.01-0.06), and Muslim women (AOR:0.29; 95%CI:0.19-0.46) were less likely to current PPFP use and intention to use contraception. Women's current use and intention to modern PPFP was found to be low with significant disparities between agrarian and pastoralist communities. Improving antenatal care, social support, women's autonomy and transforming gender equitable-norms are crucial facilitators for PPFP use. Culturally-tailored interventions are required to promote women's autonomy to enhance use of PPFP.

## Introduction

Family Planning (FP) is the practice of using contraception to attain the desired number of children and timing of pregnancies [1]. It is a crucial intervention in ending preventable maternal deaths due to pregnancy and childbirth [2]; and enables women to make informed decisions about their reproductive health [3]. Postpartum Family Planning (PPFP) is the use within 12 months after childbirth which plays a proven in protecting against unintended pregnancy and is a highly recommended intervention [1], and reduced maternal, perinatal and child deaths by preventing the risk of closely spaced pregnancies [4]. In Low- and Middle-Income Countries (LMICs), over half of the 1.6 billion women of reproductive age have the desire to control and delay their pregnancy [5]. However, 218 million women had unmet needs to access modern contraception. In addition, approximately half of the pregnancies that occur every year in LMICs are unintended [5].

Existing evidence has shown that 4.5 million women lack access to modern contraception in Ethiopia [6,7], and PPFP use remains low that ranges from 25% [6] to 45.4% [7,8]. Huge disparities between regional states in Ethiopia, 12.3% in Somali [9], to 55% in Oromia [10]. Gender dynamics are pivotal in shaping women's access to and use of PPFP in Ethiopia [11]. Gender discrimination limits access to quality family planning services for both women and men, with women facing greater challenges due to traditional roles, while men also experience gender-related barriers [12]. Although PPFP uses vary significantly across different geographical regions in LMICs [13]. Women still have limited access to the right FP information and services [11]. Poor partner or community support; cultural or religious opposition; patriarchal culture in Ethiopia#39;s pastoralist and agrarian communities drives gender inequalities in labor; resource access; healthcare decisions; health literacy; traditional customs; gender-based challenges; lack of contraceptive availability [11], and quality of counselling, awareness and attitude towards PPFP [14].

The FP service beneficiaries vary substantially among pastoral and agrarian regions. Access to PPFP service is more challenging for hard-to-reach settings, and mobile pastoralist communities than for agrarian and permanently settled communities [11,15,16]. In addition, existing evidence revealed that access to essential health service needs of mobile pastoralists communities have been given insufficient

attention compared to agrarian communities [17–20]. The disparity between agrarian and pastoral contexts has often been overlooked, despite being few researches explored and documented in previous studies. There remains a paucity of context-based evidence, which is crucial for addressing these gaps and informing effective policies. This study assessed women's current use of, and intention to use, postpartum contraception in the agrarian and pastoral contexts of Ethiopia.

## Methods

### Ethical consideration

Ethical clearance to conduct the study was obtained from the Ethiopian Public Health Association (EPHA) Research Ethical Review Board (Ref No: EPHA/OG/728/23). The study was conducted by the Declaration of Helsinki. Informed verbal and written consent were obtained from each study participant voluntarily to be included in the study. Informed consent from participant/legal guardian and assent from them were obtained from study participants who were younger than 18 years old and had not attended formal education. The collected data was kept confidential anonymously through the de-identification of names and other personal identifiers from the record/sheet. Parents/guardians in case of minor study participants and legally authorized representatives in case of illiterate participants.

### Study setting

Ethiopia, the second populous country in Africa, has an estimated, 120 million population with a growth rate of 2.6% [1,21]. Although the nation strives to avail access to maternal health through primary health care (2,3), the uptake of PPFP is extremely low and is characterized by a high variation in agrarian and pastoral communities. Ten Woredas from five Regional States, namely Chifra and Telalek Woredas from Afar; Degehabur, Gursum, and Gunagado Woredas from Somali Region with pastoral contexts; Seka Chekorsa and Shebe Sambo Woredas from Oromia Region; Bensa and Bona Zuria Woredas from Sidama Region and Shay Bench Woreda from Southwest Ethiopia Region which were representing the agrarian contexts were included. These sites were selected purposively due to having high home birth rates across both agrarian and pastoral contexts and data collection was conducted from August to October 2023.

### Project description

As part of the "*Improve Primary Health Care Service Delivery (IPHCSD)*" project, jointly implemented by Amref Health Africa and JSI with the support from the Gates foundation, designed a maternal and newborn health lifesaving package intervention to improve access to high-impact lifesaving interventions for vulnerable pregnant women including PPFP. It includes home-based delivery of lifesaving packages for pregnant mothers and newborns and is supplemented with a community empowerment intervention using Participatory Learning and Action (PLA) [22] strategy in agrarian and pastoral contexts. Formative research was conducted to establish baseline values and inform stakeholders to implement adaptative management of the community-based lifesaving interventions delivery project, including PPFP (Fig 1).

### Study design and participants

The community-based cross-sectional study was conducted among women who gave birth during the previous 12 months in ten selected agrarian and pastoral woredas of Ethiopia. Villages or kebeles, the lowest administrative units, are far from the woreda capital and with low utilization of health services were selected from agrarian and pastoral contexts. This baseline cross-sectional study was part of a larger community-based quasi-experimental embedded implementation research [23,24] to design and test lifesaving interventions package delivery and a PLA approach [22,25]. The interventions include promoting facility births, educating communities on birth preparedness and complication readiness, and facilitating for advance distribution. Whereas the PLA approach [22,25] enhances women's agency to address existing structural barriers to PPFP uptake.

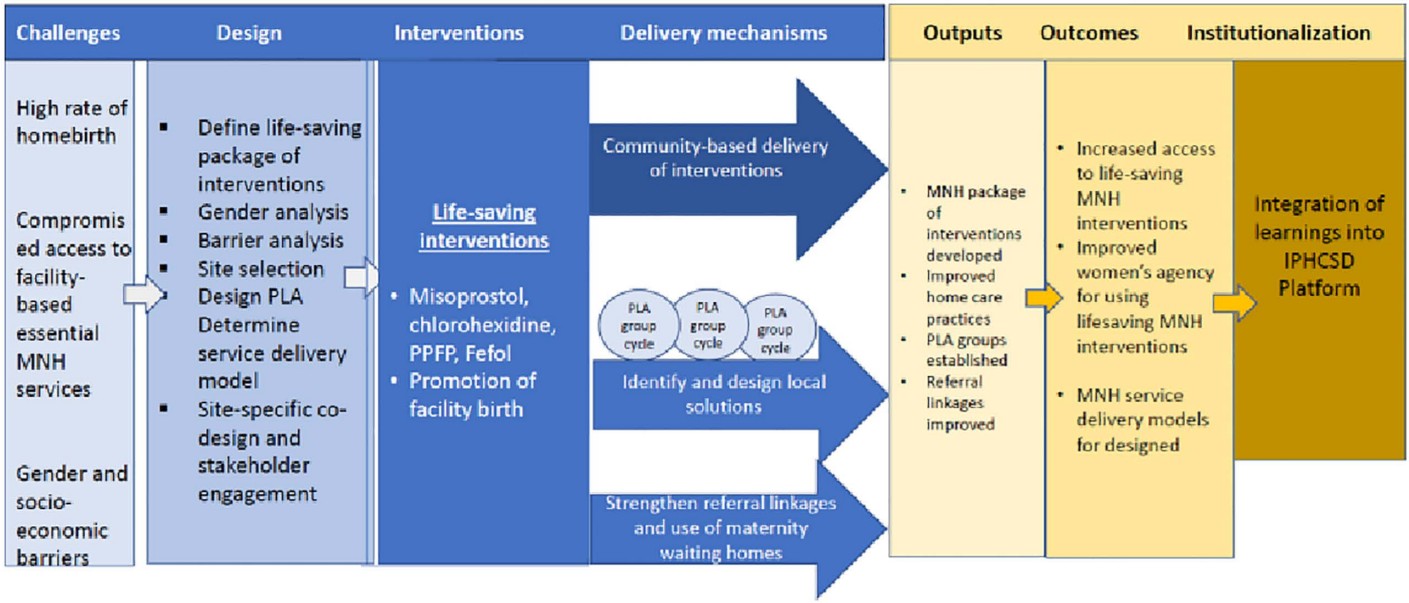

**Fig 1. Theory of change of the lifesaving package delivery project.**

## Eligibility criteria

Eligible women were residents of the 107 kebeles located in 10 selected Woredas with low facility-based maternal health service (skilled birth attendance) uptake and communities residing more than two hours walking distance from the primary health care unit, and mothers aged 15–49 years who gave birth during the previous 12 months.

## Sampling methods

The sample size was calculated using EPI Info StatCalc [26], considering the double population formula for a comparative cross-sectional household study design. The sample size calculation considered a 95% confidence level ($Z_{\alpha/2}$ = 1.96), a margin of error (d = 5%), a design effect of (e = 1.5), a power of 80%, and a two-sided significance level of 0.05 to detect 12% points change in the uptake of institutional delivery and women's decision-making autonomy on maternal health services at the end of the project; and stratified by pastoralist and agrarian settings. Accordingly, 3,097 mothers (1,242 from agrarian and 1,855 from pastoral contexts) who gave birth during the previous 12 months at the time of data collection were included in the study. A two-stage cluster sampling technique was used to select the households for in-person interviews. In the first stage, Primary Health Care Units (PHCUs) having a catchment population with a high rate of home birth were purposefully selected. There were 54 Kebeles (*Kebele is the smallest functional administrative structural unit in Ethiopia)* in the catchments of the selected PHCUs, which were the primary sampling units. The updated list of households with mothers of infants 0–11 months of age with their unique household identifiers from the electronic Community Health Information System (e-CHIS) [27] was obtained from the family folder of respective health posts, which was used as a sampling frame by the data collectors. In the second stage, women were recruited at for face-to-face interviews from their household level using a systematic random sample technique. In each sub-woreda, samples were allocated proportionally to the estimated population size and Kth interval was calculated for each PHCU's catchmen t that ranged from 4 to 16.

## Data collection

Data were collected through a house-to-house survey using a structured interview questionnaire that was customized from the Ethiopian demographic and health survey (EDHS). The questionnaire was translated into local dialects (Amharic, Afan Oromo, and Af-Somali). Experienced data collectors were recruited and trained for two days. The women#39;s interviews yielded data on sociodemographic and previous obstetric history including women's autonomy to seek maternal and newborn health (MNH) services across the continuum of care, men#39;s engagement, and access to and control over financial resources for MNH care. A web-based digital platform (SurveyCTO) was used to collect the data among women (15–49 years) who gave birth during the previous 12 months. The interviews took 45–60 minutes.

## Measurements of the outcome and exposures

The outcome variables of this study were two: current use of PPFP and Intention to use in the future. The women's current use and intention to use modern contraception during their extended postpartum period (12 months following childbirth period) was the outcome variable and was assessed based on women's self-reported binary responses (yes = 1, no = 0). The PPFP uptake was assessed by asking a woman whether "she or her partner is currently doing something or using any modern contraceptive methods to delay or avoid pregnancy during her extended postpartum period". Women's intention to use family planning was assessed by asking whether "the woman or partner will use contraceptive methods to delay or avoid pregnancy at any time in the future". The exposure variables included women's autonomy to seek MNH services, men's engagement, gender equitable attitude, and access to and control over financial resources. The gender-related barriers were assessed, including such as women's autonomy in overall decision-making [28–33], women's decision-making on family planning use, and societal attitudes towards equitable gender norms [31,34–39]. The women's attitude towards gender-equitable norms was assessed using the mean values of the response from the six items as cut-off points: '0' less than the mean value for gender-equitable and '1' above the mean for gender-inequitable norms. In addition, the social capital [40] was indirectly intended to assess the social support to the women for PPFP uses, in which social support was measured using the Oslo-3 scale [40]. Gender equitable attitude was measured using a six-item tool rated using a Likert scale ranging from 1"agree", 2"somehow agree", and 3"disagree".

## Data management and analysis

Collected data were exported from Excel Sheet to Stata/SE 18.0 [41] for cleaning and analysis. Stratified descriptive statistical analysis was used to compute the frequencies and proportions in agrarian and pastoral contexts. The characteristics of women in the two contexts were examined using Pearson's chi-square statistics test. Most of the exposure variables were categorical. Descriptive statistics were carried out to compute the proportion of exposure and outcome variables cross-tabulated with clusters. The primary health care units (PHCUs) were the clustered sampling unit, in which access to PPFP within the PHCUs catchment was similar but women's current and future PPFP use varies across PHCUs. In this paper, women's current and future PPFP use were the two outcome variables fitted independently. The community-level (cluster-level) variations were computed to assess the random-effect measure of variation. Inter-cluster correlation (ICC) and Akaike's information criterion (AIC) were estimated to assess the relationship within the cluster and variations and final model fit statistics. Eventually, multilevel mixed-effects logistic regression analysis [42,43] was fitted to determine the relationship between explanatory variables and postpartum family planning uptake and future intention stratified by agrarian and pastoral settings. Adjusted odds ratio (AOR) at 95% Confidence Interval (CI) with a p-value<0.05 was determined on the final model declaring significant association.

# Results

## Demographic and basic profiles of study participants

A total of 3,097 women were involved in this study. Women (44.3%) included in the study were in the age group of 25–30 years. The majority (97.8%) of them were ever married. Approximately three-fourths (73.4%) of the women were Muslim. Of these, 99.9% of women in the pastoral setting were Muslim. The women's sociodemographic characteristics from agrarian and pastoral settings are significantly different (Table 1).

Thirteen percent of women were primipara. Two-thirds (65.5%) of them attended antenatal care follow-up during their pregnancy period. Eighty-four (2.7%) of women experienced pregnancy within their extended postpartum period (Table 2).

## Sources of PPFP information

Approximately seven in ten (68.4%, n = 2,119) of them gave birth at home, and 60% of them received information about modern PPFP methods from either Health Extension Workers (HEWs) or other Healthcare Providers (HCP). Nearly

Table 1. Basic characteristics of women (15-49 years) in 10 Woredas from agrarian and pastoral Ethiopia, August-October 2023 (n = 3,097).

| Variables | Categories | Agrarian | Pastoral | Total | P value |
|---|---|---|---|---|---|
| Age | 15-24 | 446(35.9) | 378(20.4) | 824(26.7) | <0.001 |
| | 25-30 | 604(48.6) | 768(41.4) | 1,372(44.3) | |
| | 31-35 | 148(11.9) | 453(24.4) | 601(19.4) | |
| | 36-49 | 44(3.5) | 256(13.8) | 300(9.7) | |
| Religion | Christian | 822(66.2) | 1(0.05) | 823(26.6) | <0.001 |
| | Muslim | 419(33.8) | 1,853(99.9) | 2,272(73.4) | |
| Marital status | Ever married | 1,229(98.9) | 1,809(97.0) | 3,029(97.8) | <0.001 |
| | Never married | 13(1.1) | 55(2.9) | 68(2.2) | |
| Women education | Not attended formal school | 548(44.1) | 1,665(89.8) | 2,213(71,5) | <0.001 |
| | Primary | 346(27.8) | 117(6.3) | 463(15.0) | |
| | Secondary & plus) | 348(28.0) | 73(3.9) | 421(13.6) | |
| Husband education | Not attended formal school | 355(28.6) | 1,430(77.1) | 1,785(57.6) | <0.001 |
| | Primary | 332(26.7) | 1,148(8.0) | 480(15.5) | |
| | Secondary & plus) | 555(44,7) | 277(14,9) | 832(26.9) | |
| Women occupation | Unemployed | 1,146(92.4) | 1,672(90.4) | 2818(91.2) | <0.001 |
| | Engaged in job | 94(7.6) | 177(9.6) | 271(8.8) | |
| Husband occupation | Unemployed | 12(0.9) | 191(10.6) | 203(6.7) | <0.001 |
| | Engaged in job | 1,212(99.0) | 1,613(89.4) | 2,825(93.3) | |
| Distance from health facilities | <1 hour | 1,033(83) | 1,353(72.9) | 2,386(77.0) | <0.001 |
| | ≥ 1hours | 209(16.8) | 502(25.1) | 711(27.9) | |
| Family size | ≤5 members | 532(42.8) | 382(20.6) | 914(29.51) | <0.001 |
| | >5 Members | 710(57.2) | 1,473(79.4) | 2,183(70.5) | |
| Wealth index | Poor | 65(5.2) | 968(52.2) | 1,033(33.4) | <0.001 |
| | Middle | 340(27.4) | 692(37.3) | 1,032(33.3) | |
| | Rich | 837(67.4) | 195(10.5) | 1,032(33.3) | |
| Social support | Poor | 284(22.9) | 417(22.5) | 701(22.6) | |
| | Moderate | 478(38.5) | 1,176(63.4) | 1,654(53.4) | |
| | Strong | 480(38.7) | 262(14.1) | 742(23.9) | |
| Membership to VHL | Yes | 168(13.5) | 50(2.7) | 218(7.0) | <0.001 |
| Membership to CBHI | Yes | 691(55.6) | 17(0.9) | 708(22.9) | <0.001 |

**Table 2. Women's obstetric history in the agrarian and pastoral settings, Aug-Oct, 2023 (n = 3,097).**

| Variables | Categories | Agrarian | Pastoral | Total | P value |
|---|---|---|---|---|---|
| Gravida q36a | primigravida | 261(21.0) | 137(7.4) | 398(12.9) | <0.001 |
| | multigravida | 981(78.9) | 1,718(92.6) | 2,699(87.1) | |
| Para q36b | Primipara | 261(21.0) | 137(7.4) | 398(12.9) | <0.001 |
| | Parity 2–5 | 650(52.3) | 1,198(64.6) | 1,848 (59.7) | |
| | Grand multipara>5 | 331(26.7) | 520(28.0) | 851(27.5) | |
| Antenatal care | Had follow up | 1,093(88.0) | 934(50.4) | 2,017(65.5) | <0.001 |
| Place of birth | Home birth | 502(40.4) | 1,617(87.2) | 2,119(68.4) | <0.001 |
| | Facility birth | 740(59.6) | 238(12.8) | 978(31.6) | |
| Pregnancy status | Yes | 3(0.2) | 81(4.4) | 84(2.7) | <0.001 |
| | No | 12291(99.0) | 1,674(90.2) | 2,903(93.7) | |
| | Not sure | 10(0.8) | 100(5.4) | 110(5.6) | |

one in five (19.7%, n = 609) women discussed with the HEW or other HCPs about PPFP. Of these, 99.7% (607) of them obtained necessary information about birth control methods to be taken within 45 days of the postpartum period. Most of the women were told about injectables (86.5%), implants (71%), and pills (59.9%) to avoid pregnancy during postpartum period. Nevertheless, discussion about contraceptives with HCPs was extremely low with significant variation in agrarian (43.5%, n = 540) and pastoral (3.7%, n = 69) contexts.

### Postpartum FP uptake

The majority of the women (97.3%, n = 3,013) of them were interviewed about either themselves or their partner currently using any modern FP methods to delay or prevent pregnancy. Of these, women's postpartum contraceptives use was 25.3% (95%CI:23.8-26.9%) with significant disparity between agrarian and pastoral contexts (**Fig 2**).

Additionally, among women who used PPFP (n = 762), only 2.5% (n = 19) of women used PPFP immediately within 48 hours of gave birth. Nearly three-quarters (71.3%, n = 543) of them started PPFP within six weeks (**Fig 3**).

### Preferred PPFP methods

Of those women who gave birth in the past 12 months, approximately three-fourths (73.1%, n = 557) of them used injectable methods. Although it has low uptake in pastoral contexts, overall injectables and implants were preferred methods (**Table 3**).

### Sources of PPFP service

Among the women studied, the majority accessed PPFP methods primarily through health centers, with health posts being the next most common access point (**Fig 4**). Of those who used PPFP, 98.7% (n = 752) received their preferred method. Only a small proportion (1.3%, n = 10) did not obtain their methods of choice, mainly reasons such as method stock out, lack of trained skilled personnel, and/or the absence at the specific facility during women's visits for PPFP.

### Reasons for not using PPFP

Women had several reasons for not using postpartum contraceptives. The majority reported religious prohibition (40%), being lactating (21.1%) and a desire to have more children (18.1%) as the primary factors influencing the women's decisionmaking to use PPFP (**Table 4**).

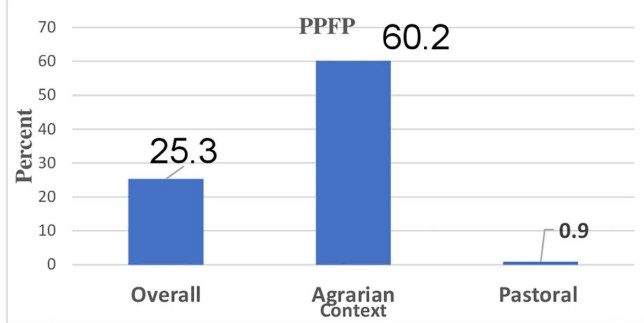

**Fig 2. Women's current use of PPFP in the agrarian and pastoral Ethiopia, Aug-Oct 2023 (n = 3,013).**

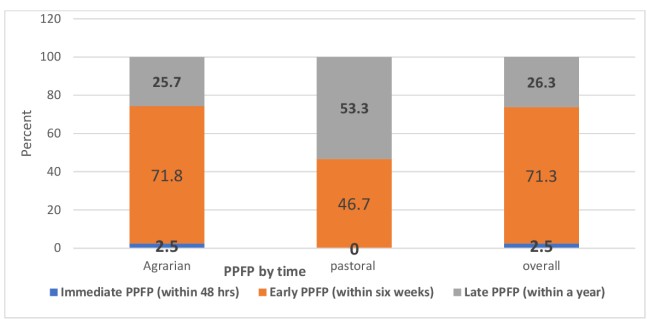

**Fig 3. Women's PPFP use by period in the agrarian and pastoral settings of Ethiopia, Aug-Oct, 2023 (n = 762).**

**Table 3. Women's preferred contraceptive choices for PPFP in agrarian and pastoral Ethiopia (n = 762).**

| Family planning methods | Total | Agrarian | Pastoral |
|---|---|---|---|
| Injectable | 557(73.1) | 553(74.0) | 4(26.7) |
| Implants | 175(23.0) | 171(22.9) | 4(26.7) |
| Pill | 17(2.2) | 15(2.0) | 2(13.3) |
| IUD | 3(0.4) | 3(0.4) | 0 |
| Others* | 10(0.01) | 5(0.01) | 5(0.3) |

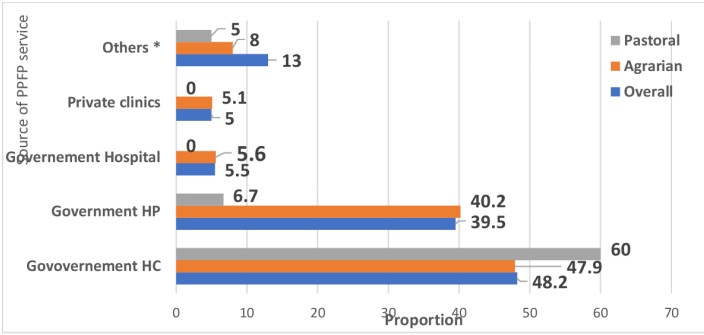

**Fig 4. Sources of postpartum family planning methods in the agrarian and pastoral settings of Ethiopia, Aug-Oct 2023.**

**Table 4. Women's common reasons for not using PPFP in the agrarian and pastoral context of Ethiopia, Aug-Oct, 2023 (n = 3097).**

| Reason for not using PPFP Variable | Total | Agrarian | Pastoral |
|---|---|---|---|
| Religious prohibition | 905(40.3) | 19(3.9) | 886(50.4) |
| Breastfeeding | 474(21.1) | 147(30.3) | 327(18.6) |
| Want to get pregnant | 406(18.1) | 22(4.5) | 384(21.8) |
| Husband/partners opposed | 215(9.6) | 42(8.6) | 173(9.8) |
| Postpartum amenorrhoeic | 198(8.8) | 181(37.2) | 17(1.0) |
| Knows no method | 113(5.0) | 6(1.2) | 107(6.1) |
| Method of choice not available | 106(4.7) | 8(1.7) | 98(5.6) |
| Women opposed contraceptive | 94(4.2) | 10(2.1) | 84(4.8) |
| Not have sex | 92(4.1) | 68(14.0) | 24(1.4) |
| Other family members opposed | 86(3.8) | 0(0.0) | 86(4.9) |
| Fear of side effects | 75(3.4) | 9(1.9) | 66(3.8) |
| Know no source | 62(2.8) | 1(0.2) | 61(3.5) |
| Lack of access/far | 32(1.4) | 16(3.3) | 16(0.9) |
| Health concern | 18(0.8) | 4(0.8) | 14(0.8) |
| Infrequent sex | 18(0.8) | 9(1.9) | 9(0.5) |
| Cost too much | 16(0.7) | 4(0.8) | 12(0.7) |
| Inconvenient to use | 7(0.3) | 2(0.4) | 5(0.3) |

## Women's intention to use contraception

Women's future intention was assessed among only those who were not using PPFP during the study period. The women's intention to use contraception was assessed, overall slightly more than one-third (37.5% 95%CI:35.8-39.2%) of women expressed intention to use PPFP in the future. Nevertheless, only 5.4% of women who live in pastoral settings had an intention to use PPFP in the future (**Fig 5**). Injectable (57.1%), the implant (34.9%) and pills (3%) were the preferred methods by women for future use.

## Factors determining the women's use of PPFP

A total of sixteen explanatory variables were considered in the mixed effect logistic regression analysis. Women from Muslim communities (AOR:0.29; 95%CI:0.19-0.46); living more than an hour#39;s walking distance from the health facilities (AOR:0.70; 95%CI:0.49-0.99); residing in pastoral contexts (AOR:0.03; 95%CI:0.01-0.06); women's previous history of home birth (AOR:0.53; 95%CI:0.39-0.72) and women who had autonomy to jointly decide on place of delivery (AOR:0.38; 95%CI:0.22-0.66)

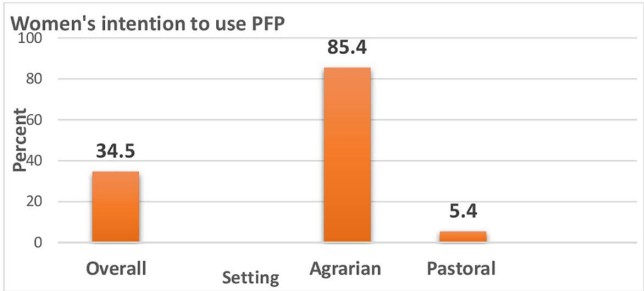

**Fig 5. Women's intention to use contraception in the agrarian and pastoral Ethiopia, Aug-Oct, 2023.**

were significantly associated barriers to PPFP use of women in agrarian and pastoral contexts of Ethiopia. Women who had strong social support (AOR:1.75; 95%CI:1.23-2.49); had the autonomy to decide on FP use: 3.25; 95%CI:1.89-5.59) and had antenatal care follow-up (AOR:3.46; 95%CI:2.25-5.32) were significantly associated with women's uptake of PPFP in Ethiopia (**Table 5**). The null model of the regression analysis considers the inter-cluster variability between PHCUs as a cluster (**Table 6**).

## Factors determining women's intention to use contraception

A total of sixteen explanatory variables were considered in the mixed effect logistic regression analysis to identify independent predictors of women's intention to use PPFP in the future. Of these, Women who attended secondary school and above (AOR:1.87; 95% CI:1.14-3.04); wealth index (women from rich groups) (AOR:1,76; 95% CI:1.07-2.89); women who lived in families who have favorable attitudes towards equitable gender norms (AOR:1.64; 95% CI:1.19-2.26); women who had antenatal care (AOR:1.92; 95% CI:1.35-2.74); women who had strong social support (AOR:2.01 95% CI:1.34-3.02) and women who have the autonomy to use FP (AOR: 2.57; 95% CI:1.61-4.10) were the factors significantly associated with the positive intention of women to use PPFP in their subsequent births. Nevertheless, women from pastoral context (AOR:0.07; 95% CI:0.04-0.16); Muslim women (AOR:0.14; 95% CI:0.08-0.27); women who gave their previous birth at home (AOR: 0.35; 95% CI:0.25-0.49); and live far from the health facility (more than an hour walking distance) (AOR: 0.62; 95% CI: 0.61-0.93) were less likely to use the PPFP in the future in Ethiopia (**Table 7**). The null model of the regression analysis considers the inter-cluster variability between PHCUs as a cluster (Table 8).

**Table 5. Multilevel mixed-effects logistic regression model of the factors affecting women's uptake of PPFP in Ethiopia, Aug-Oct 2023.**

| Variables | Categories | (COR 95% CI) | Model 2: Intercept model AOR (95% CI) |
|---|---|---|---|
| Age of women (ref: 15–24) | 23-30 | 0.84(0.64-1.10) | 1.01(0.72-1.40) |
| | 31-35 | 0.70(0.47-1.05) | 0.92(0.57-1.50) |
| | ≥36 | 0.69(0.37-1.30) | 0.84(0.41-1.72) |
| Maternal education (ref: not attended school) | Primary | 1.32(0.97-1.80) | 0.94(0.68-1.33) |
| | Secondary | 2.28(1.64-3.18) | 1.43(0.97-2.11) |
| Religion (ref: Christian) | Muslim | 0.10(0.05-0.23) | 0.29(0.19-0.46)* |
| Husband education (ref: not attended school) | Primary | 1.47(1.05-2.08) | 1.22(0.85-1.74) |
| | Secondary+ | 1.79(1.29-2.49) | 1.08(0.74-1.57) |
| Wealth index (ref: poor) | Middle | 1.46(0.87-2.43) | 1.04(0.60-1.79) |
| | Rich | 1.92(1.16-3.22) | 0.99(0.57-1.72) |
| Distance to HF (ref: <1hour walking distance) | ≥1 hour | 0.72(0.51-1.03) | 0.70(0.49-0.99)* |
| Gender-equitable attitude (ref: No) | Yes | 1.62(1.23-2.13) | 1.51(1.12-1.71)* |
| Family size (ref: <5 family members) | ≥5 members | 0.63(0.49-0.81) | 0.79(0.57-1.10) |
| Social support (ref: poor) | Moderate | 1.29(0.93-0.1.78) | 1.35(0.96-1.89) |
| | Strong | 1.60(1.15-2.23) | 1.75(1.23-2.49)* |
| Being member of VHL (ref: No) | Yes | 1.58(1.05-2.39) | 1.29(0.85-1.95) |
| Being member of CBHI (ref: No) | yes | 1.31(0.98-1.74) | 1.00(0.74-1.35) |
| Autonomy to seek FP (ref: No) | Yes | 2.51(1.74-3.62) | 3.25(1.89-5.59)* |
| ANC follow up (ref: No | yes | 4.07(2.70-6.16) | 3.46(2.25-5.32)* |
| Place of delivery (ref: Health Facility) | Home birth | 0.42(0.30-0.57) | 0.53(0.39-0.72)* |
| Autonomy to delivery place (ref: Husband) | Jointly or alone | 1.23(0.85-1.78) | 0.38(0.22-0.66)* |
| Residence/community livelihood (ref: agrarian) | Pastoral | 0.004(0.002-0.01) | 0.03(0.01-0.06)* |

*NB: *significant association at p-values <0.05; HF: health facility; ANC: antenatal care; FP: family planning.*

**Table 6. Random effects (measures of variations) and model fitness for the factors affecting women's uptake of PPFP in Ethiopia: Multilevel mixed-effects logistic regression modeling, Aug-Oct 2023.**

| Parameters | Null model (empty model) | Final model |
|---|---|---|
| Community-level intercepts (SE) | -1061.757 | -786.578 |
| **Random-effect (measures of variation)** | | |
| Cluster level variance (SE) | 14.644(3.337) | 0.231(0.125) |
| Loglikelihood (LL) | -928.267 | -7779.555 |
| **Model fit statistics** | | |
| ICC (SE) | 0.0.817(0.034) | 0.066(0.331) |
| AIC(BIC) | 1860.534(1872.534) | 1607.109(1751.35) |

AIC: Akaike's information criterion, DIC: Deviance information criterion, ICC: Intra-class correlation coefficient, SE: Standard error.

**Table 7. Multilevel mixed-effects logistic regression model of the factors affecting women's intention to use contraception in Ethiopia, Aug-Oct 2023.**

| Variables | Categories | COR 95% CI | Intercept model AOR (95% CI) |
|---|---|---|---|
| **Age of women (ref: 15–24 year)** | 25-30 years | 0.71(0.52-0.96) | 0.65(0.45-0.95)* |
| | 31-35 years | 0.49(0.32-0.73) | 0.49(0.29-0.80)* |
| | ≥36 years | 0.53(0.31-0.94) | 0.47(0.25-0.89)* |
| Maternal education (ref: not attended school) | Primary | 1.82(1.26-2.63) | 1.19(0.79-1.77) |
| | Secondary+ | 3.13(2.02-4.84) | 1.87(1.14-3.04)* |
| Religion (ref: Christian) | Muslim | 0.01(0.004-0.03) | 0.14(0.08-0.27)* |
| Husband education (ref: not attended school) | Primary | 1.46(1.01-2.11) | 1.13(0.77-1.68) |
| | Secondary+ | 1.99(1.42-2.81) | 1.10(0.74-1.62) |
| Wealth (ref: Poor) | Middle | 2.02(1.31-3.13) | 1.40(0.89-2.19) |
| | Rich | 3.84(2.37-6.22) | 1,76(1.07-2.89)* |
| Distance to HF (ref: walking hours<1) | ≥1 hour | 0.74(0.48-0.1.08) | 0.62(0.61-0.93)* |
| Gender-equitable attitude (ref: No) | Yes | 1.75(1.30-2.36) | 1.64(1.19-2.26)* |
| Autonomy to HF delivery (ref: husband) | Alone or joint | 1.75(1.25-2.46) | 0.69(0.41-1.12) |
| Place of delivery (ref: HF) | Home birth | 0.22(0.16-0.31) | 0.35(0.25-0.49)* |
| ANC follow (ref: No) | Yes | 2.81(2.03-3.92) | 1.92(1.35-2.74)* |
| Family size (ref: family size <5) | ≥5 members | 0.81(0.61-0.1.08) | 1.39(0.96-2.02) |
| Social support (ref: poor) | Moderate | 0.1.36(0.96-1.93) | 1.33(0.95-1.92) |
| | Strong | 2.11(1.44-3.08) | 2.01(1.34-3.02)* |
| Autonomy to seek FP (ref: Husband) | Joint or alone | 261(1.87-3.62) | 2.57(1.61-4.10)* |
| Member of VHL (ref: No) | Yes | 1.43(0.81-2.52) | 1.26(0.84-1.90) |
| Member of CBHI (ref: No) | Yes | 1.73(1.19-2.52) | 0.99(0.55-1.77) |
| Residence/community livelihood | Agrarian | 1.0 | 1.00 |
| | Pastoral | 0.005(0.002-0.01) | 0.07(0.04-0.16* |

*NB: *significant association at p-values <0.05; HF: health facility; ANC: antenatal care; FP: family planning.*

**Table 8. Random effects (measures of variations) and model fitness for the factors affecting women's intention to use PPFP in Ethiopia: Multilevel mixed-effects logistic regression modeling, Aug-Oct 2023.**

| Parameters | Null model (empty model) | Final model |
|---|---|---|
| Community-level intercepts (SE) | -1063.232 | -741.214 |
| **Random-effect (measures of variation)** | | |
| Cluster level variance (SE) | 12.659(2.479) | 0.555(0.197) |
| Loglikelihood (LL) | -913.031 | -0.741.214 |
| **Model fit statistics** | | |
| ICC (SE) | 0.794(0.032) | 0.1444(0.044) |
| AIC(BIC) | 1830.61(1842.138) | 1530.429(1675.33) |

AIC: Akaike's information criterion, DIC: Deviance information criterion, ICC: Intra-class correlation coefficient, SE: Standard error.

## Discussion

This study highlights that one-fourth of women have ever used PPFP, with significant differences among women of reproductive age who live in the agrarian and pastoral contexts in Ethiopia. Though some women give birth at health facilities, most missed the opportunity to receive PPFP services within 48 hours of the postpartum period. Factors such as women who had previous antenatal care visits, strong social support, autonomy on FP use, and living in a community who have a favorable attitude towards equitable gender norms were crucial facilitators for the use of PPFP. Nevertheless, inaccessibility to health facilities, history of home birth, being a pastoral community, and being Muslim were some of the barriers to women using PPFP in Ethiopia. Additionally, religious prohibitions further discouraged women from using PPFP.

The level of PPFP use reported in this study was consistent with the study's findings in Malawi [44], Uganda [10,30,45], Pakistan [46,47], and Ethiopia [48,49]. Nevertheless, there are studies with higher prevalence of PPFP use in Ethiopia [50–52], and other agrarian settings of Ethiopia [53]. Furthermore, some studies reported the lowest (12.3%) use of PPFFP in the Ethiopia-Somali Region [54,55]. In the meantime, more than one-third (37.5%) of women expressed an intention to use PPFP, with a notably higher rate of 84.5% in agrarian communities compared to only 5.4% in pastoral communities. This finding is consistent with reported women's intentions in Benin [56]. These substantial discrepancies call for action to contextualize PPFP services suitable to pastoral communities. Similarly, there is low-level use of PPFP in Liberia (11.9%) [57], the Democratic Republic of Congo [58], and Ethiopia as well [59] which underscores the challenges in expanding FP services. This finding highlights the need to promote diverse contraceptive options.

The mixed-effect multilevel analysis reveals the key determinants of PPFP use in the study area, including religious background, access to healthcare service, attitudes toward gender-equitable norms, social support, and women's autonomy on health service use and PPFP uptake. Similarly, research in Liberia found an inverse relationship between distance and PPFP use [57]. A previous study in Ethiopia [60] reported higher odds of contraceptive use among women who have access to health facilities [57]. In addition, women who live in communities with favorable attitudes towards equitable gender norms were more likely to adopt PPFP [61]. Empowered women in gender-equal societies make autonomous decisions on contraception. Furthermore, socio-cultural norms, cultural practices like polygamy, and the preference for large family members negatively influenced women's use of modern contraception [62]. Similarly, women's contraceptive use was affected by their partner's knowledge, and beliefs which was reported in SSA countries [62]. The existence of strong social support particularly for spouses increases the likelihood of the use of PPFP in Ethiopia [48], Indonesia [61], Kenya [63], and Nigeria [37]. Similarly, women who have autonomy in their health-seeking including FP services more likely to

 

use PPFP. These evidences promote women's empowerment as a key intervention in reproductive health decision-making in some SSA countries [56,64,65], and Indonesia [61].

In this study, the use of PPFP was discouraged by the presence of religious prohibitions and traditional beliefs. This finding is consistent with existing literature [66] and is supported by qualitative studies from Burkina Faso [67] and Sierra Leone [68] where religious leaders often equate contraception with abortion and female sterilization, while emphasizing birth spacing. Furthermore, religious leaders in these contexts lack adequate training on contraception and have limited or no engagement with sexual and reproductive health services. These factors have significant implications for Ethiopia's PPFP program, particularly in multicultural and religiously diverse settings such as pastoralist areas, which are predominantly Muslim.

### Strengths and limitations of this study

The study used primary data collected from randomly selected women. A community-based study covers multiple sites and with statistically estimated sample size for each enumeration area. The livelihood of women from agrarian and pastoral settings was considered to accommodate cultural, norms and contextual variations towards PPFP use. Furthermore, a digital platform (computer-assisted personal interviews (CAPI) was applied for data collection and prompt data quality tracking. The questionnaire was adapted from the Ethiopian Demographic and Health Survey protocol which is standardized and validated for similar purposes. Nevertheless, the accuracy of the data depends on the women's self-report response and inquiries in the past 12 months period might be affected by a recall basis. Although disaggregated analysis by setting: agrarian versus pastoral was planned and carried out for descriptive analysis section, limited uptake of PPFP methods in the pastoral setting restricted us to perform a separate model for inferential statistics to the agrarian and pastoralist contexts.

### Implication of the study

In Ethiopia, women are entrenched with many barriers affecting them in seeking health service, and disproportionally high in the pastoral contexts. In response to this, the IPHC-SD project conducted this baseline assessment to re-orient maternal health service delivery platforms to improve access to quality and impactful lifesaving maternal and neonatal health package intervention delivery within the existing primary health care system of Ethiopia. Hence, understanding the level of PPFP use and women's intention to use contraception in the future is crucial to designing strategies to improve PPFP use. The findings highlight the huge disparity between women who live in the agrarian and pastoral pettings of Ethiopia which has an immense contribution to inform project designing to address the barriers using participatory learning and action methodology.

### Conclusions

The PPFP use remains low in the study settings with pronounced disparities between agrarian and pastoral communities in Ethiopia. The findings highlight the necessity for context-tailored interventions that address the distinct challenges faced by women in pastoral settings. Key determinants influencing current PPFP uptake and future intentions include women's age, religion, attitude toward gender-equitable norms, women's autonomy for family planning use, antenatal care use, and strong social support play pivotal roles in influencing both the current uptake and intention to use PPFP. This finding is informative for designing culture-sensitive and women-friendly community-based delivery of high-impact lifesaving maternal and neonatal health packages including PPFP options, across diverse in Ethiopian settings.

### Supporting information

**S1 File. Dataset.**

(XLSX)

## Acknowledgments

We would like to acknowledge the Gates Foundation, Amref Health Africa in Ethiopia, and JSI for the overall administrative support, the study participants, data collectors, and supervisors for their willingness to give their time and information for this study. We would like to extend our gratitude to the project staff, namely, Ms. Zehara Fenataw, Mr. Hassen Musse, Ms. Rediet Daniel, and Mr. Abdellah Mohammed for their coordination for the data acquisition process.

## Author contributions

**Conceptualization:** Agumasie Semahegn, Gizachew Tadele Tiruneh, Alemnesh Hailemariam Mirkuzie, Nebreed Fesseha, Addis Girma, Chala Tesfaye, Hillina Tadesse, Derbe Tadesse Abate, Netsanet Belete, Lidiya Tefera, Frank DelPizzo, Abdulhalik Workicho, Muluken Dessalegn Muluneh, Temesgen Ayehu, Dessalew Emaway, Misrak Makonnen, Mesele Damte Argaw.

**Data curation:** Agumasie Semahegn, Gizachew Tadele Tiruneh, Omar Mohammed, Shegaw Mulu, Wubegzier Mekonnen, Biruk Bogale, Meskerem Abebaw, Mebrie Belete, Miftah Yasin, Netsanet Belete.

**Formal analysis:** Agumasie Semahegn, Gizachew Tadele Tiruneh, Wubegzier Mekonnen.

**Funding acquisition:** Gizachew Tadele Tiruneh, Alemnesh Hailemariam Mirkuzie, Nebreed Fesseha, Addis Girma, Hillina Tadesse, Lidiya Tefera, Abdulhalik Workicho, Muluken Dessalegn Muluneh, Addis Tamire, Temesgen Ayehu, Dessalew Emaway, Misrak Makonnen.

**Investigation:** Agumasie Semahegn, Gizachew Tadele Tiruneh, Alemnesh Hailemariam Mirkuzie, Nebreed Fesseha, Mikiyas Teferi, Derbe Tadesse Abate, Mesele Damte Argaw.

**Methodology:** Agumasie Semahegn, Gizachew Tadele Tiruneh, Alemnesh Hailemariam Mirkuzie, Nebreed Fesseha, Shegaw Mulu, Wubegzier Mekonnen, Chala Tesfaye, Mikiyas Teferi, Biruk Bogale, Derbe Tadesse Abate, Netsanet Belete, Yibeltal Kifle Alemayehu, Abdulhalik Workicho, Mesele Damte Argaw.

**Project administration:** Agumasie Semahegn, Alemnesh Hailemariam Mirkuzie, Omar Mohammed, Nebreed Fesseha, Shegaw Mulu, Addis Girma, Hillina Tadesse, Meskerem Abebaw, Miftah Yasin, Netsanet Belete, Zemzem Mohammed, Lidiya Tefera, Frank DelPizzo.

**Resources:** Alemnesh Hailemariam Mirkuzie, Mebrie Belete, Lidiya Tefera, Muluken Dessalegn Muluneh, Addis Tamire.

**Software:** Agumasie Semahegn, Gizachew Tadele Tiruneh.

**Supervision:** Agumasie Semahegn, Alemnesh Hailemariam Mirkuzie, Omar Mohammed, Nebreed Fesseha, Shegaw Mulu, Addis Girma, Mikiyas Teferi, Biruk Bogale, Hillina Tadesse, Derbe Tadesse Abate, Meskerem Abebaw, Mebrie Belete, Miftah Yasin, Netsanet Belete, Zemzem Mohammed, Yibeltal Kifle Alemayehu, Abdulhalik Workicho, Muluken Dessalegn Muluneh.

**Validation:** Agumasie Semahegn, Gizachew Tadele Tiruneh, Alemnesh Hailemariam Mirkuzie, Omar Mohammed, Nebreed Fesseha, Shegaw Mulu, Wubegzier Mekonnen, Addis Girma, Chala Tesfaye, Mikiyas Teferi, Biruk Bogale, Hillina Tadesse, Derbe Tadesse Abate, Meskerem Abebaw, Netsanet Belete, Zemzem Mohammed, Lidiya Tefera, Frank DelPizzo, Yibeltal Kifle Alemayehu, Abdulhalik Workicho, Muluken Dessalegn Muluneh, Addis Tamire, Temesgen Ayehu, Dessalew Emaway, Misrak Makonnen, Mesele Damte Argaw.

**Visualization:** Agumasie Semahegn, Gizachew Tadele Tiruneh, Shegaw Mulu, Frank DelPizzo, Mesele Damte Argaw.

**Writing – original draft:** Agumasie Semahegn.

**Writing – review & editing:** Agumasie Semahegn, Gizachew Tadele Tiruneh, Alemnesh Hailemariam Mirkuzie, Omar Mohammed, Nebreed Fesseha, Shegaw Mulu, Wubegzier Mekonnen, Addis Girma, Chala Tesfaye, Mikiyas Teferi, Biruk Bogale, Hillina Tadesse, Derbe Tadesse Abate, Meskerem Abebaw, Mebrie Belete, Miftah Yasin, Netsanet

Belete, Zemzem Mohammed, Lidiya Tefera, Frank DelPizzo, Yibeltal Kifle Alemayehu, Abdulhalik Workicho, Muluken Dessalegn Muluneh, Addis Tamire, Temesgen Ayehu, Dessalew Emaway, Misrak Makonnen, Mesele Damte Argaw.

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
