## [Decision Letter · Decision Letter 0]

3 Mar 2026

PGPH-D-25-04096

Women’s postpartum family planning use and intention in Ethiopia: Disparities in the agrarian and pastoral contexts from community-based cross-sectional study

Dear Dr. Semahegn,

Thank you for submitting your manuscript to PLOS Global Public Health. After careful consideration, we feel that it has merit but does not fully meet PLOS Global Public Health’s publication criteria as it currently stands. Therefore, we invite you to submit a revised version of the manuscript that addresses the points raised during the review process.

We look forward to receiving your revised manuscript.

Kind regards,

Damen Haile Mariam, MD, MPH, PhD

Academic Editor

Journal Requirements:

1. Please provide a detailed online Financial Disclosure statement. This is published with the article. It must therefore be completed in full sentences and contain the exact wording you wish to be published.

a) State the initials, alongside each funding source, of each author to receive each grant. For example: “This work was supported by the National Institutes of Health (####### to AM; ###### to CJ) and the National Science Foundation (###### to AM).”

For more information, please go to our submission guidelines:

https://journals.plos.org/globalpublichealth/s/submission-guidelines#loc-financial-disclosure-statement

2. Please ensure that the funders and grant numbers match between the Financial Disclosure field and the Funding Information tab in your submission form. Note that the funders must be provided in the same order in both places as well.

3. In the online submission form, you indicated that "All related data are presented fully within the paper, and available upon reasonable request to the lead author and the corresponding author.".

a) In a public repository,

b) Within the manuscript itself, or

d) Uploaded as supplementary information.

For further assistance, you may go to: http://journals.plos.org/globalpublichealth/s/data-availability

4. Please provide separate main figure files in .tif or .eps format only and ensure that all files are under our size limit of 10MB.

5. Please include a separate legend or caption for each figure in your main manuscript.

6. Please upload a copy of "Supplement file 1", which you refer to in your text on page 6 and ensure that it has a legend listed in the manuscript before or after the references list. Or if the supplementary material is no longer to be included as part of the submission, please remove all reference to it within the text.

Additional Editor Comments (if provided):

Reviewer 1 -

- Abstract/introduction:

- There is a description of Immediate PPFP- since the study looked at primarily on the extended period of PPFP and very limited descriptions on IPPFP, it is better to remove the word Immediate. In Ethiopia, Immediate Postpartum Family Planning

(IPPFP) refers to providing family planning counseling and methods to women within 48 hours after childbirth, before they leave the health facility, to prevent unintended pregnancies and promote birth spacing, integrating services with early

postpartum care for better maternal and child health outcomes. It involves offering a wide range of safe contraceptive options like implants, injectables, pills, IUDs (PPIUD), and permanent methods, addressing unmet needs and reducing risks

associated with short interpregnancy.

- Line-2 in the introduction- the word CONTROL is very controversial and the reference document you put gives a better wording on the use of contraception. "Family Planning allows people to attain their desired number of children and determine

the spacing of pregnancies".

- Line 24 in the introduction. beyond health care decision, the quality of health care is a very critical issue which is not described. Since most delivering mothers are discharged without a contraceptive method, one should ask on the quality of

post-partum care. This might be a useful reference to look at it- Silesh M, Demisse TL, Taye BT, Moltot T, Chekole MS, Wogie G, Kasahun F, Adanew S. Immediate postpartum family planning utilization and its associated factors among

postpartum women in Ethiopia: a systematic review and meta-analysis. Front Glob Women's Health. 2023 Aug 22;4:1095804. doi: 10.3389/fgwh.2023.1095804. PMID: 37674902; PMCID: PMC10478094.

- Methods:

- Line-2 in the project description- Figure-2 is a theory of change not PPFP service distribution. Please check it again.

- Results:

- Line-19 under the results section- Post Partum uptake- 2.5% of the participants has immediate PPFP within 48 hours and it is good to include on how many of them had facility births.

- Table-3 title needs more clarity- Since preferred choice is not always actual use and you need to revise the title to represent actual use. The table represents the actual contraceptive use.

- Under results line 6-8 there seems a calculation error- with 762 users of which 752 use their preferred choice which should be 98.7% and the remaining 10 in like 1.3% without their preferred choice. Please correct the error.

- On table-4 on reasons of not using PPFP, is the religious prohibition a real one or an individual perception? Since a large percentage of the respondents fall under that category it is good to describe that in both the results and discussion section.

- Discussion:

- The are a number of findings in the study and I expect more to be described in the discussion section. As of now it is limited with three paragraphs with primarily comparing with other studies and very limited justification on some of the results.

As an example, I don't see any description on the reason of having almost a 2.5% use of Immediate PPFP within 48 hours, while almost close to 32% delivered at facilities and most left without contraception. It is good to look at some of the

justification which can be used for a call to action for anyone using the results of such studies.

- In few areas there is a need for word editing as an example Recall bias is written as Recall Basis, pastoral setting as Pastoral petting.

Reviewer 2 -

- Introduction:

- In the statemen of the problem section, the problem reflected a real researchable gap or challenge. But there are some confusing sentences to be considered like line numbers 32 and 33: 32 is stated as: "contexts has not been recognized but

is well explored and comprehensively documented through previous studies.’ These sentences seem to contradict one another and have clarity and logic problems. As written, it seems the wo ideas cancel each other out.

- Methods:

- The methods section starts by stating the ethical issues, which is not the main focus of the section. It has to come at the end of the section. It has to follow this sequence: Study setting, study design, population, sample and sampling

techniques, method of data analysis, data presentation, eligibility criteria, and ethical clearance.

- In the "Study design and participants" sub-section, there is a redundancy in numbers 6 & 7 and numbers 10-12.

Reviewers' comments:

Reviewer's Responses to Questions

**Comments to the Author**

1. Does this manuscript meet PLOS Global Public Health’s publication criteria? Is the manuscript technically sound, and do the data support the conclusions? The manuscript must describe methodologically and ethically rigorous research with conclusions that are appropriately drawn based on the data presented.? Is the manuscript technically sound, and do the data support the conclusions? The manuscript must describe methodologically and ethically rigorous research with conclusions that are appropriately drawn based on the data presented.

Reviewer #1: Yes

2. Has the statistical analysis been performed appropriately and rigorously?

Reviewer #1: Yes

3. Have the authors made all data underlying the findings in their manuscript fully available (please refer to the Data Availability Statement at the start of the manuscript PDF file)?

The PLOS Data policy requires authors to make all data underlying the findings described in their manuscript fully available without restriction, with rare exception. The data should be provided as part of the manuscript or its supporting information, or deposited to a public repository. For example, in addition to summary statistics, the data points behind means, medians and variance measures should be available. If there are restrictions on publicly sharing data—e.g. participant privacy or use of data from a third party—those must be specified.requires authors to make all data underlying the findings described in their manuscript fully available without restriction, with rare exception. The data should be provided as part of the manuscript or its supporting information, or deposited to a public repository. For example, in addition to summary statistics, the data points behind means, medians and variance measures should be available. If there are restrictions on publicly sharing data—e.g. participant privacy or use of data from a third party—those must be specified.

Reviewer #1: Yes

4. Is the manuscript presented in an intelligible fashion and written in standard English?

Reviewer #1: Yes

Reviewer #1: The authors looked at a very critical area of intervention in preventing maternal and child mortality and the study was done in hard to reach places to understand the use of PPFP which can be generalized further to similar areas. I have the following few general feedback which will help the manuscript to go out in a better shape.

1. At the start of the abstract/introduction there is a description of Immediate PPFP- since the study looked at primarily on the extended period of PPFP and very limited descriptions on IPPFP, it is better to remove the word Immediate. In Ethiopia, Immediate Postpartum Family Planning (IPPFP) refers to providing family planning counseling and methods to women within 48 hours after childbirth, before they leave the health facility, to prevent unintended pregnancies and promote birth spacing, integrating services with early postpartum care for better maternal and child health outcomes. It involves offering a wide range of safe contraceptive options like implants, injectables, pills, IUDs (PPIUD), and permanent methods, addressing unmet needs and reducing risks associated with short interpregnancy.

2. Line-2 in the introduction- the word CONTROL is very controversial and the reference document you put gives a better wording on the use of contraception. "Family Planning allows people to attain their desired number of children and determine the spacing of pregnancies".

3. Line 24 in the introduction. beyond health care decision, the quality of health care is a very critical issue which is not described. Since most delivering mothers are discharged without a contraceptive method, one should ask on the quality of of post-partum care. This might be a useful reference to look at it- Silesh M, Demisse TL, Taye BT, Moltot T, Chekole MS, Wogie G, Kasahun F, Adanew S. Immediate postpartum family planning utilization and its associated factors among postpartum women in Ethiopia: a systematic review and meta-analysis. Front Glob Women's Health. 2023 Aug 22;4:1095804. doi: 10.3389/fgwh.2023.1095804. PMID: 37674902; PMCID: PMC10478094.

4. Line-2 in the project description- Figure-2 is a theory of change not PPFP service distribution. Please check it again.

5. Line-19 under the results section- Post Partum uptake- 2.5% of the participants has immediate PPFP within 48 hours and it is good to include on how many of them had facility births.

6. Table-3 title needs more clarity- Since preferred choice is not always actual use and you need to revise the title to represent actual use. The table represents the actual contraceptive use.

7. Under results line 6-8 there seems a calculation error- with 762 users of which 752 use their preferred choice which should be 98.7% and the remaining 10 in like 1.3% without their preferred choice. Please correct the error.

8. On table-4 on reasons of not using PPFP, is the religious prohibition a real one or an individual perception? Since a large percentage of the respondents fall under that category it is good to describe that in both the results and discussion section.

9. The are a number of findings in the study and I expect more to be described in the discussion section. As of now it is limited with three paragraphs with primarily comparing with other studies and very limited justification on some of the results. As an example I don't see any description on the reason of having almost a 2.5% use of Immediate PPFP within 48 hours, while almost close to 32% delivered at facilities and most left without contraception. It is good to look at some of the justification which can be used for a call to action for anyone using the results of such studies.

10. In few areas there is a need for word editing as an example Recall bias is written as Recall Basis, pastoral setting as Pastoral petting.

**Do you want your identity to be public for this peer review?** For information about this choice, including consent withdrawal, please see our Privacy Policy..

Reviewer #1: **Yes:** Mengistu Asnake Kibret (MD, MPH)Mengistu Asnake Kibret (MD, MPH)Mengistu Asnake Kibret (MD, MPH)Mengistu Asnake Kibret (MD, MPH)

Independent Global Public Health Expert

---

## [Editor Report · Decision Letter 1]

15 Mar 2026

PGPH-D-25-04096R1

Women’s postpartum family planning use and intention in Ethiopia: Disparities in the agrarian and pastoral contexts from community-based cross-sectional study

Dear Dr. Semahegn,

Thank you for submitting your manuscript to PLOS Global Public Health. After careful consideration, we feel that it has merit but does not fully meet PLOS Global Public Health’s publication criteria as it currently stands. Therefore, we invite you to submit a revised version of the manuscript that addresses the points raised during the review process.

We look forward to receiving your revised manuscript.

Kind regards,

Damen Haile Mariam, MD, MPH, PhD

Academic Editor

**Journal Requirements:**

**Additional Editor Comments (if provided):**

Minor Comments:

- Most of the comments have been addressed. However, there are still some minor errors that should be taken care of such as:

- Methods section - The last sentence of the first paragraph in the "project description" sub-section - Figure 1 should be "Theory of Change", and not "distribution of PPFP services".

- Results section - There is discrepancy of numbers (762 Vs 782) in the sentence just above table 4 that says - "Among women who used PPFP, 98.7% (n=752) of them obtained their preferred method of choice. The remaining 4% (31) of women did not get the preferred method".

- Editorial issues - There are a number of minor editorial issues that need to be taken care of such as the second sentence under the "Reasons for not using PPFP" sub section that says - "The major of the women reported religious prohibition...".
---

## [Editor Report · Decision Letter 2]

19 Mar 2026

Women’s postpartum family planning use and intention in Ethiopia: Disparities in the agrarian and pastoral contexts from community-based cross-sectional study

PGPH-D-25-04096R2

Dear Dr. Semahegn,

We are pleased to inform you that your manuscript 'Women’s postpartum family planning use and intention in Ethiopia: Disparities in the agrarian and pastoral contexts from community-based cross-sectional study' has been provisionally accepted for publication in PLOS Global Public Health.

Best regards,

Damen Haile Mariam, MD, MPH, PhD

Academic Editor